# A Review of the Relationship between Socioeconomic Status Change and Health

**DOI:** 10.3390/ijerph20136249

**Published:** 2023-06-29

**Authors:** Caroline Barakat, Theodore Konstantinidis

**Affiliations:** Faculty of Health Sciences, Ontario Tech University, 2000 Simcoe St. N, Oshawa, ON L1G 0C5, Canada

**Keywords:** socioeconomic status, social mobility, social determinants of health, health outcomes, life course, social inequalities, health resiliency, cumulative health effects

## Abstract

Objectives: This review aimed to identify and synthesize the existing literature on the effects of socioeconomic status (SES) changes on health. Methods: A review was conducted using Medline, Cochrane library, and CINAHL (Cumulative Index to Nursing and Allied Health Literature). All longitudinal or cross-sectional studies that examined links between changes to SES across different time periods and measured health outcomes were included. Screening was conducted using select inclusion and exclusion criteria in order of title, abstract, and full text. Two independent reviewers assessed the quality of the full text articles using the Downs and Black checklist. Results: Our literature search led to 2719 peer reviewed articles, 2639 of which were title screened after duplicates were removed. A total of 117 abstracts and 12 full text articles were screened. Overall, findings from 11 articles form the basis of this review. Eight different types of measures of changes to SES were identified. These include education, occupation, economic security, income sufficiency, home ownership, car ownership, health insurance, and marital status. Assessed outcomes included measures related to physical health, cardiovascular disease, mental health, and oral health. A large proportion of studies found that an SES change impacts health. Evidence suggests that those with consistently high SES have the best health outcomes, followed by those who report their SES change from low to high (upward social mobility). Evidence on the relative health effects for those who report their SES change from high to low (downward social mobility) compared to those who report consistently low SES is inconsistent. Conclusion: Current evidence suggests that an SES change has an impact on an individual’s health. More research on the effects of SES changes on health outcomes in adulthood is needed and can inform various areas of health research including health resiliency and development. Future studies should focus on individual SES indicators and their effects on health outcomes at multiple points throughout life.

## 1. Introduction

Public health is a broad field that aims to prevent disease, prolong life, and promote health [1]. Socioeconomic status (SES) is one of the key determinants of health that refers to the social class or position of an individual or group. It is commonly measured using income, education, and occupation [2]. Evidence shows that SES is related to health outcomes with those who experience a relatively low SES having a lower life expectancy and an increase in chronic conditions when compared to those of a higher SES [3]. There is a consensus of strong correlations between income and health, housing and health, and education and health [4]. For instance, lower levels of education were associated with a higher risk of cardiovascular disease [5]. Indeed, some of the most studied and well-correlated health outcomes in relation to socioeconomic status are cardiovascular health, diabetes, life expectancy, obesity, and mental health [6].

Using a life-course approach to examine relationships between socioeconomic status and health outcomes often pours into one of three hypotheses: the social mobility hypothesis, the critical period hypothesis, and the accumulation hypothesis [7]. The social mobility hypothesis postulates that changes in socioeconomic status throughout life have an influence on health outcomes [8]. The social mobility hypothesis focuses on measures that indicate social position, such as income, education, occupation, and housing. It postulates that social mobility is inversely linked to income inequality. A greater gap in income inequality leads to low social mobility [8]. The critical period hypothesis postulates that there is a specific point in an individual’s life where societal factors influence health outcomes throughout life [7]. Particularly, this hypothesis focuses on earlier socioeconomic status in an individual’s life and their influence on health outcomes in later life. For instance, individuals who are classified as belonging to a higher SES early in life benefit from opportunities that facilitate optimal childhood development, including optimum physical growth, good oral health, coping mechanisms, and resilience, all leading to better physical and mental health outcomes later in life. The accumulation hypothesis postulates that the effects of advantaged or disadvantaged socioeconomic status accumulate over the life course, ultimately affecting an individual’s health. The accumulation hypothesis focuses on accumulation across socioeconomic status categories at various points throughout an individual’s life which contribute to health outcomes.

The social mobility hypothesis infers that individuals of a relatively lower SES are more likely to remain in this category [8]. By the same token, those of relatively higher SES who have greater wealth, education, and opportunities, are more likely to accumulate related benefits across their life course, such as better access to health care, good quality housing, and food security, all contributing to better health outcomes. The critical period hypothesis perspective holds that children who belong to relatively lower SES households are likely to be exposed to higher pollution environments, have low access to health care, and generally be subject to harmful factors during critical periods, which negatively impact their health in the long-term [7]. On the other hand, the accumulation perspective suggests that individuals who experience a relatively high SES throughout their life course are most likely to have better health while those who experience a relatively low SES throughout their life course are most likely to experience negative health outcomes [7].

However, several questions arise. Which of these hypotheses best explains the link between an SES change and diverse health outcomes? What happens if social mobility takes its course, and individuals move from a high social position to a low social position or vice versa? How is health affected? How do health effects of moving from a high-to-low social position differ from moving from a low-to-high social position? Examining relationships between SES and health using a life course approach necessitates data on different time periods, which also allows an assessment of an SES change and its impact on health.

This review examines the existing literature on the impacts of SES changes on health over time. While strong consistent evidence on links between SES and health exists, research on SES changes and health is understudied and can inform diverse areas of research, such as health resiliency, health development, immigrant health, environmental health, and public health. Specifically, how does an SES change impact health? How is health impacted by an increase in SES (upward social mobility), a decrease in SES (downward social mobility), consistently low SES, and consistently high SES? Are the health effects of SES cumulative across the life course?

## 2. Methods

A literature search was conducted following the principles outlined by the Cochrane Handbook of Systematic Reviews. The databases included in the search were Medline, Cochrane Library, and the Cumulative Index to Nursing and Allied Health Literature (CINAHL) for all research published by 13 December 2019. These databases are commonly used for medical and health-related research reviews. Since this was a scoping review, the use of these databases ensured the inclusion of comprehensive multi-disciplinary coverage, peer-reviewed original articles, systematic reviews, and meta-analyses, which facilitated the mapping of available evidence on the relationship between SES change and health. In all databases, terms related to “socioeconomic,” “education,” “occupation,” “income,” “housing,” “health,” and “change” were combined. Figure 1 presents a detailed search strategy used in Medline that was adapted for use in Cochrane Library and CINAHL.

Inclusion and exclusion criteria were applied to identify appropriate studies that examined links between SES change and measured health outcomes. Inclusion criteria consisted of 4 points: (i) participant data on SES represented a period of at least 10 years; (ii) SES change was measured through changes to income, education, occupation, health insurance, housing, or any socioeconomic factor; (iii) at least one health outcome was measured at baseline and at follow up; and (iv) longitudinal study designs or cross-sectional designs that allow for measurement of SES change over time as well as at least one health outcome. Articles were screened using a multi-level process by title and abstract and followed with full screening. At each stage of the screening, articles that did not meet the inclusion criteria were removed. In preparation for data extraction, all articles were quality assessed using the Downs and Black checklist. Two independent reviewers assessed study quality to ensure transparency and methodological quality with the intention to exclude studies that scored less than 15. A data extraction sheet was used to collect information related to year of publication, research study design, study location, sample population, sample size, research questions, socioeconomic status measured, health outcome measured, and main findings. This information was synthesized and tabulated. Findings were grouped on the basis of health dimension of interest, such as physical health, mental health, or respiratory health.

## 3. Results

The initial search yielded 2719 peer reviewed articles. After the duplicate articles were removed, 2639 articles remained for title screening. Title screening led to the removal of 2438 articles that did not focus on SES changes and 84 articles that did not focus on at least one health outcome. This led to abstract screening for 117 articles, of which 105 were excluded due to an absence of specific measures of SES changes (n = 61) and due to research designs other than longitudinal or cross-sectional (n = 44). Finally, 12 articles were full text screened, and 1 was excluded as the article was not written in English. In total, 11 articles fulfilled the inclusion criteria and were included in this review (Figure 2). Eight main types of measures for change to SES were identified: education, occupation, economic security, income, home ownership, car ownership, health insurance, and household class/occupational prestige [9,10,11,12,13,14,15,16,17,18,19]. While most articles measured education using the individual’s education level, a few focused on maternal or paternal education [11,14,18]. The same was seen for occupation, with more articles focusing on paternal occupation [12,13,17] rather than maternal. Economic security was assessed using measures related to income [9,14,15], income sources [15], ‘times of hunger’ [9], delivery payment mode [14], deprivation [18], and having financial problems paying bills [17]. Income was measured using classifications derived from continuous measures of family income [9,14], mean income across time [15], and income changes [14,15]. Home ownership related to one’s ownership as well as parental home ownership [11,17]. Other measures included Hollingshead Index (which assessed social status using multiple factors related to occupation and education), occupation social class, household class, and parental occupational class [10]. All articles used at least two of these types of measures. Examined health outcomes can be broadly placed into 4 categories: (1) general physical health, (2) cardiovascular disease, (3) mental health, and (4) oral health. General physical health was assessed in the included articles through examinations of various physical health outcomes [10], self-rated heath, Body Mass Index (BMI) [13,16], frailty [9], arthritis [11], and allostatic load [12]. Cardiovascular disease was assessed using general cardiovascular health [18], diagnosis with hypertension [16], and cardiovascular-related mortality [17]. Mental health was assessed using prevalence of mental disorders, suicide attempts, alcohol misuse, life purpose, self-acceptance, personal growth, environmental mastery, and psychological well-being [14,15]. Oral health was assessed using the severity of dental caries and mean dental caries [19]. Table 1 provides a summary of the data extracted including assessed exposures and outcomes as well as the main findings. Various associations were found between SES changes and each of the categories of the examined health outcomes.

(i)SES change and general physical health

In relation to general physical health, two articles examined links between upward social mobility and physical health or frailty in adulthood [9,10]. Alvarado et al. (2008) focused on a large sample of men and women (n = 10,661) over the age of 60 who had participated in the SABE project (Salud Bienerstar y Envejecimiento) [9]. They found that those who had impoverished childhoods were more likely to have higher rates of frailty in later life. Those who also had hardship in later life, relatively low education, and worked in a manual occupation were even more likely to be frail beyond 60 years of age than their counterparts [10]. Cundiff et al. (2017) found that upward social mobility is protective for physical health, independent of baseline (childhood) or current SES, race, physical childhood health, and child or adolescent BMI [10]. The researchers focused on 1165 participants who were recruited from a pool of children registered in Grade 1 and followed every 6 months for four years and then annually for nine years. The research found that the correlation of upward social mobility on physical health burden in adulthood was a strong protective factor at all three stages in later life (middle childhood, late childhood, and middle adolescence) and had an additive protective effect over time.

Two studies found that a decrease in social mobility predicted worse health outcomes in adulthood than a consistent low SES [11,12]. While examining the link between SES changes and arthritis for 1276 participants recruited through 22 family practice clinics in North Carolina, Baldassari et al. found that the relationship between SES and self-reported diagnosis with arthritis was greatest in those whose social mobility went from a medium to low SES (OR 2.20, 95% Confidence Interval (CI), 1.29–3.75), followed by a low to low SES (OR 2.05, 95% CI, 1.35–3.12), and lastly, a low to medium SES (OR 1.76, 95% CI, 1.15–2.70) [11]. Similar findings are reported from a study based on Scotland where researchers sampled three different sub-cohorts of differing ages (15, 35, & 55 years of age) across five data collection points starting 1987 to 2008 [12]. Based on a total of 2580 participants, this study found that participants who experienced a higher socioeconomic position during all life stages had the lowest mean allostatic load. Highest allostatic loads were seen in individuals who had a relatively higher SES in childhood followed by a relatively lower SES in the remaining life stages. Researchers concluded that these findings are better explained by the accumulation model than by the social mobility model. While the previously mentioned studies focused on the health benefits of social mobility, Padyab et al. (2014) rejected the social mobility hypothesis and supported the accumulation and critical period models when examining links between SES changes and body mass index for women [13]. Specifically, this research found that women who were in the lower socioeconomic position had a significantly higher BMI compared to those who were continuously in the higher socioeconomic position. Moreover, BMI was positively associated with the socioeconomic score for women across three time periods, suggesting support to the accumulation of risk model for women. The research did not find evidence for life course associations between SES and BMI among men.

(ii)SES change and mental health

Focusing on mental health, a Brazil-based study found that the prevalence of mental disorders by age 30 increased as the SES decreased [14]. Findings are based on a cohort of 3701 participants from the 1982 Pelotas Birth Cohort who were followed-up in 2012 in order to examine associations between mental disorders at age 30 years and gender, socioeconomic position at birth, and family income trajectories. Those who were always poor (22.4% *p* < 0.001) had the highest prevalence of mental health disorders followed by non-poor then poor (19.0% *p* < 0.001), poor then non-poor (12.4% *p* < 0.001), and never poor (9.6% *p* < 0.001) [13]. These findings agree with an earlier unrelated US-based study that focused on the cumulative impact of different income measures on psychological well-being among 1127 adults using a life course approach [15]. Participants were sampled across four different time periods in 1965, 1974, 1983, and 1994. Using diverse measures of mean income, income slope, and income sources, the study found that as income increased over time, so did their purpose in life (*β* = 0.250, *p* < 0.0001), self-acceptance (*β* = 0.223, *p* < 0.001), personal growth (*β* = 0.113, *p* < 0.001), environmental mastery (*β* = 0.134, *p* < 0.001), and autonomy (*β* = 0.061, *p* < 0.069). The study also notes that the increase in psychological well-being is linked with the quantity and frequency of profit income.

(iii)SES changes and cardiovascular health

Three reviewed articles focused on the relationship between SES changes and cardiovascular disease (CVD) [16,17,18]. A US-based study that focused on three groups of youth and adolescents-formerly in foster care, economically insecure, and economically secure–at baseline found that risk factors for cardiovascular disease (BMI and hypertension) consistently increased in odds for youths who were classified as economically secure (ES) to those who are economically insecure (EI) (OR 1.65 (*p* < 0.001)) to the foster care group [16]. Another study focused on the links between risk factors and current SES using longitudinal data from the GLOBE study in Netherlands for a sample of 27,027 participants [17]. The main objective was to contribute to knowledge on the underlying socioeconomic inequalities in cardiovascular mortality in adulthood. The study found that the direct contribution of childhood risk factors was 8% for those with a consistently low SES, 12% for those whose social mobility went from a middle to low SES, and 7% for those who moved from a middle to high SES while the direct contribution of adulthood risk factors was 45% for those who experienced a consistently low SES, 40% for decreased social mobility (middle to low SES), and 10% for increased social mobility (middle to high SES). Lastly, a study on over 300 twin pairs found differences in systolic and diastolic blood pressure in monozygotic twins who were ‘discordant’ in adulthood socioeconomic status in relation to occupation type [18]. The twin who was classified as working-class was more likely to have a higher systolic and diastolic blood pressure than the non-working-class twin. Furthermore, the twin who experienced cumulative deprivation had the worst health in adulthood, demonstrating that the health benefits of SES accumulate across the life course.

(iv)SES change and oral health

One reviewed study that examined the relationship between SES change and oral health found that that the severity of dental caries-measured by mean DFS (Decayed and Filled primary dental Surfaces)-was lowest in those whose SES changed from low to high (10.09 *p* < 0.001) and the highest in those whose SES remained low (12.38, *p* < 0.001) [19]. When examining mean DS (Decayed Surfaces), the frequency of dental caries increased as follows: SES change classified as high to high (1.26, *p* < 0.001), low to high (1.61, *p* < 0.001), high to low (1.94, *p* < 0.001), and low to low (2.05, *p* < 0.001).

## 4. Discussion

This paper examined evidence on links between SES changes and health. While there is evidence that an SES change is directly linked to health status, it remains unclear whether experiencing a decrease in SES (high to low) leads to poorer physical health outcomes in adulthood compared to those who experience a consistently low SES (low to low) [9,10,11,12]. For instance, two reviewed studies that focused on frailty or medical diagnoses found that a consistently low SES led to the most detrimental health outcomes while two reviewed studies that focused on self-reported diagnosis with arthritis or allostatic load report that downward social mobility is linked to the worst health outcomes. These differences may be due to several factors, including differences in assessing SES change, the outcome of interest, or geographically specific or population-specific factors that mediate the link between SES changes and health.

In relation to mental health, the literature findings point to a clearer link between SES changes and health outcomes, where upwards social mobility is related to improved mental health outcomes. For instance, evidence suggests that individuals who were never poor had the lowest prevalence of mental health disorders followed in order by those who moved up the SES gradient, those who moved down the SES gradient, and those who were always poor having the highest prevalence of mental health disorders [14]. By the same token, higher average income and increasing income were positively associated with a higher purpose in life, self-acceptance, personal growth, and environmental mastery (mental health) [15]. These findings are consistent with existing research on the positive influence of socioeconomic status (SES) on mental health outcomes [20].

In relation to cardiovascular health, individuals who experienced a relatively higher SES measured by education, income, occupation, home and car ownership, and financial security fared better than their counterparts [16,17,18]. Those who were economically secure had fewer cardiovascular risk factors than those who were economically insecure while the latter group had fewer cardiovascular risk factors than those who were placed in foster care during childhood [16]. When adjusting for childhood and adulthood socioeconomic status, adulthood socioeconomic status has a significantly greater effect on cardiovascular mortality [17]. Even in monozygotic twins, the working-class twin was more likely to have higher blood pressure than the non-working-class twin [18]. Those who experienced greater cumulative decreases in socioeconomic status had an increased risk for cardiovascular disease. Similar to the literature findings on the link between SES changes and mental health, the link between SES changes and cardiovascular health appears to be unambiguous. This is not surprising considering that mental health and cardiovascular health are intricately linked [21], and both appear to be impacted by downward social mobility, though the timing of exposure and outcome cannot be ascertained from this review. In relation to oral health, both frequency and severity of dental caries impacts individuals who are on the trajectory of continuous low socioeconomic status [19].

An examination of evidence related to SES changes and BMI suggests that the negatively associated relationship between SES and BMI is gender-based. This is consistent with previous study that supports associations between life course SES and obesity for women [22]. While it is possible that BMI for women is more impacted by SES change than that for men, specific SES factors (occupation, education, or income) as well as diverse compositional (race, age) and contextual factors (such as neighborhood characteristics and whether one resides in developed versus developing countries) may have varied and inconsistent influence on BMI than on other health outcomes [23].

Our findings in relation to the links between SES change and each of the four examined health dimensions support the need for public health policies that focus on socioeconomically disadvantaged individuals with an aim to improve their physical and mental health, and their wellbeing. While traditional interventions that improve access to healthcare and increase health awareness are important, policies that address inequalities in education, income, and occupation are integral to promoting health equity. By addressing these fundamental issues, the effect of socioeconomic disparities on health can be lessened and overall population health can improve. Although traditional interventions such as enhancing mental health awareness and expanding mental health service accessibility benefit population health, our results suggest that addressing socioeconomic disparities and mitigating impacts of SES change are key factors in improving the overall mental health of communities. Consequently, it may be prudent to address SES disparities when designing public health policies that focus on community mental health. Our results that focus on the relationship between SES changes and cardiovascular health add to evidence that highlights the importance of addressing socioeconomic inequalities for public health. Addressing these socioeconomic disparities may be most effective in improving cardiovascular health, particularly given the evidence of the role that other key and SES-linked social determinants of health—such as gender, race, and ethnicity—play in impacting cardiovascular health [24]. Focused social policies that promote equality in terms of education, employment, financial stability, gender, ethnicity, and other social determinants of health have the capacity to benefit the health of underprivileged individuals and reduce cardiovascular health disparities. Oral health is another dimension that can benefit from social policies that focus on reducing socioeconomic disparities. Given research that links oral health with overall health [25] and reviewed evidence that oral health is positively associated with SES [19], addressing SES disparities has the capacity to improve oral health, which in turn will lead to improved population health outcomes.

When examining relationships between SES changes over time and health outcomes, it may be that the social mobility model, the critical period hypothesis, and the accumulation hypothesis work individually or in conjunction to explain diverse health outcomes. Overall, most of the reviewed studies (8 out of 11) point to a direct relationship between SES change and health outcomes in adulthood. That is, consistent high SES and upward social mobility (low to high) give rise to better health outcomes in later life than consistently low SES and downward social mobility (high to low) [9,10]. Moreover, 2 of the 11 studies found that downward social mobility (high to low) gives rise to worse health outcomes than experiencing consistently low SES [11,12]. It is worth noting that the social mobility hypothesis was rejected when examining the relationship between an SES change and BMI [13]. This was based on the belief that the social mobility model sees all upward trends as equally beneficial and all downward trends as equally harmful. Furthermore, if there is no social mobility, this would then imply that there is no change in BMI. However, none of these hypotheses were supported as BMI changed even when socioeconomic status stayed stagnant, and all trends were not equal [13]. Lastly, the accumulation hypothesis was supported by one study that found a greater accumulation of SES over time leads to a lower allostatic load [12]. The critical period hypothesis was explored in the reviewed studies but was not used to explain relationships between SES changes and health.

## 5. Limitations

This review was subject to limitations related to logistics and time. For instance, we limited our search to studies published in English and those of longitudinal or cross-sectional designs with a follow-up period greater than 10 years. A more detailed systematic review may lead to more contributions to knowledge in this area. Our review findings cannot be overgeneralized since they are based on limited research, relatively small sample sizes, and localized geographic research. Specifically, patterns of racism and social exclusion are geographically dependent and contribute to persistently low social mobility, including limited education and occupational opportunities, housing, income disparities, access to health care, psychological and emotional impacts, and intergenerational impact. Lastly, the diversity in definitions and measures of SES changes and health outcomes presents limitations in comparing research findings and assessing the weight of the evidence.

## 6. Conclusions and Future Research

Social mobility appears to impact health. In general, evidence suggests that individuals who experience consistently high SES and those who experience upward social mobility have better health outcomes throughout life and in adulthood than those who experience consistently low SES and downward social mobility. However, focusing on the latter group, it is unclear as to whether poorer health outcomes are more correlated with consistently low SES or downward social mobility, whether the positive health effects of relatively high SES dissipate over time with downward social mobility, and whether health resiliency is indirectly linked with SES and plays a role in protecting individuals from downward social mobility. Future research on this topic can benefit from consistent and diverse measures of SES changes and health outcomes, longer follow-up periods, regular multi-point data collection, and the inclusion of ethnically and geographically diverse cohorts with large sample sizes to ensure representation in each of the social mobility groups. Importantly, future research can benefit from testing policies and programs that promote improved wellbeing on various health outcomes over time.

## Figures and Tables

**Figure 1 ijerph-20-06249-f001:**
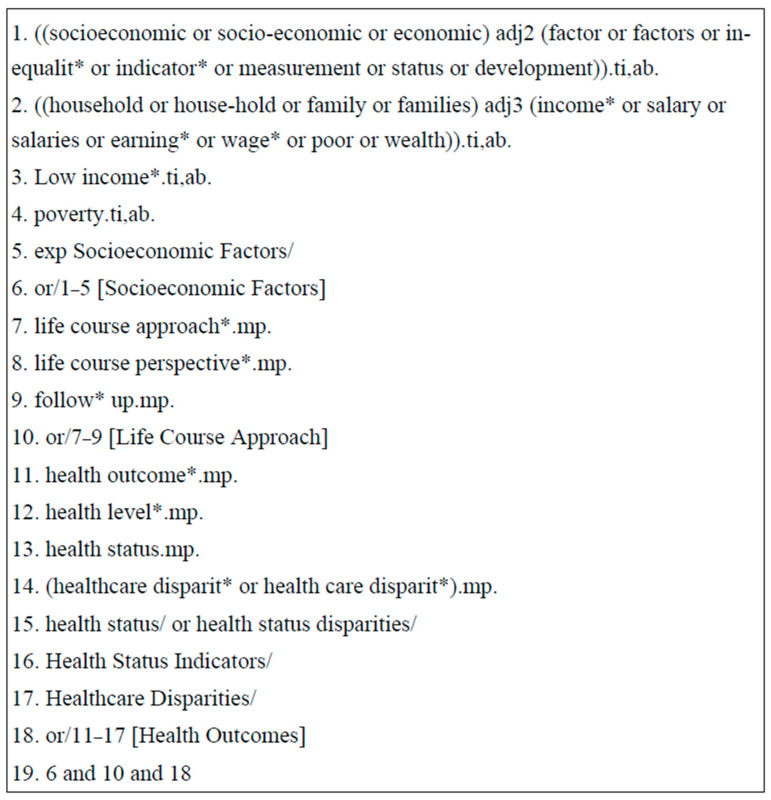
Ovid Medline Search Strategy to examine links between socioeconomic change and health across time. Note: This search strategy was created in Ovid Medline and adapted for use in Cochrane Library and CINAHL.

**Figure 2 ijerph-20-06249-f002:**
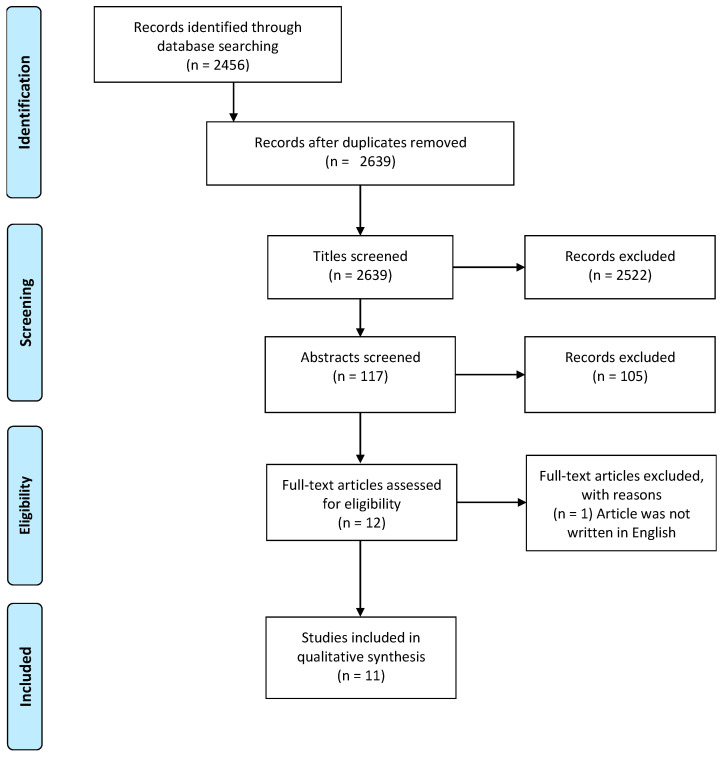
PRISMA Flow Diagram leading to the inclusion of eleven studies in this review.

**Table 1 ijerph-20-06249-t001:** Summary of Data on the links between Socioeconomic status change and health.

Author, Year	Study Design *	Country	n	Time Period	Exposure & Outcome Measures	Main Findings
Ahrens et al., 2014 [16]	P	USA	2513 youths & adolescents	17/18 to 25/26 years	Education, economic insecurity, physical health	Consistent pattern of increased health risk, BMI, and hypertension in relation to change from economically secure to insecure.
Alvarado et al., 2008 [9]	C	Seven Latin American and Caribbean (LAC) cities	10,661 men and women over the age of 60 years	Retrospective data for participants over the age of 60 years	Gender, family economics, times of hunger, education, occupation, income, marital status & self-rated health	Those who had experienced impoverished childhoods were more likely to be frail. Those of low education, manual occupation, and perceived economic hardship later in life were related to greater likelihood of frailty after 60 years of age.
Baldassari et al., 2013 [11]	C	USA	1276	Retrospective data for participants between 22 and 94 years (mean = 57 years)	Age, ethnicity, gender, BMI, self and parental homeownership and occupation, education & Arthritis	The relationship between SES and arthritis was greatest in groups that fell from middle to low SES, followed by always low, and lastly low to middle SES. The risk of developing arthritis was 2 times greater for individuals who fell from high to low SES but not statistically significant due to small sample size.
Barros et al., 2018 [14]	R	Brazil	3701	30-year follow-up	Family income, delivery payment mode, mother’s schooling, height, skin color & Mental health	Low SES was linked with mental disorders, suicide attempts, and alcohol misuse. Never poor had the lowest prevalence. Individuals whose families started their life as poor but then improved their SES by age 30 had lower prevalence than those who started off in a better SES and became poor.
Cundiff et al., 2017 [10]	P	USA	1165 males	Students attending first grade were assessed every 6 months for the first 4 years and then annually for 9 years	Hollingshead Index (marital status, retired/employed status, educational, and occupational prestige) & Physical health	Upward mobility of SES has a protective effect on physical health, independent of baseline childhood SES, current adult SES, race, physical childhood health, and child and adolescent BMI. Increase in SES has an additive protective effect over time.
Kamphuis et al., 2012 [17]	R	Netherlands	27,027	17-year follow-up	Education, occupation of father, type of health insurance, car ownership, housing tenure, and financial problems with paying bills & Physical health	When adjusted, adulthood SES had significantly greater contribution to cardiovascular disease (CVD) mortality independent of childhood SES. Most of the association with childhood SES was its contribution to adulthood risk factors. Those with greater adulthood material had a lower risk of CVD mortality.
Kaplan et al., 2008 [15]	R	USA	1127	Four data collection periods between 1965 and 1999	Mean income, Income slope, and income sources & Mental health	Average income, income increases, and receipt of profit income were positively associated with higher purpose in life, self-acceptance, personal growth, and environmental mastery (mental health). Profit income positively associated with psychological well-being
Krieger et al., 2005 [18]	R	USA	352 twin pairs	0–14 years	Adult household class, father’s education level, adult education level & Physical health	Poorer health was more likely to be reported in the working-class twin. Cardiovascular health differed among discordant twins based on their adulthood socioeconomic position. Those who experienced cumulative deprivation had the worst health.
Padyab & Norberg, 2014 [13]	R	Sweden	3440	60+ years	Education, and Occupational class & Physical health (BMI)	The social mobility model was rejected when looking at SES and BMI. Findings supported the accumulation and critical period models of SES and BMI for women
Robertson et al., 2014 [12]	R	Scotland	2580	20-year period	Head of household occupational social class at age 15 & adulthood, highest education attained & Physical health	Those who had higher socioeconomic position (SEP) during all 3 examined life-stages had the lowest mean allostatic load. Highest allostatic load was seen in individuals who had a higher childhood SEP followed by lower SEP in the 2 remaining life-stages. The greater SEP accumulation over time the lower the allostatic load.
Thomson et al., 2004 [19]	P	New Zealand	1037	Assessments conducted at ages 0, 3, 5, and 26 years	Parental occupational class, adult occupational class & Physical health	The severity of dental caries (mean Decayed and Filled primary dental Surfaces (DFS)) was lowest in the low to high SES trajectory group and highest in the low-to-low group. Mean dental caries were lowest to highest as follows: high to high, low to high, high to low, low to low. Upward mobility hypothesis is supported as those who rose from low to high SES had the best oral health outcomes second to those that were high to high. Downward mobility was also supported as high to low had the worse oral health second to low to low SES.

* P—prospective cohort; R—retrospective cohort; C—cross-sectional.

## Data Availability

Not applicable.

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
