# Peer review of "A Review of the Relationship between Socioeconomic Status Change and Health"

_ijerph, 2023, doi:10.3390/ijerph20136249_

Round 1
Reviewer 1 Report
The aim of this was to identify and synthesize existing literature on the effects of socioeconomic status change on health. Three databases were used (Medline, Cochrane library and CINAHL). The review included 11 articles, six prospective cohorts, two cross sectional studies and two prospective cohorts. The Downs and Black checklist were used for quality check. The scope of this paper is interesting and very intriguing; however, the review is short and limited. Some part of the article must be clarified, and some questions needs to be raised. The results section needs an uplift, and the results should be clearer written.
Please clarify the literature search. The search periods. The search period ended December 13th 2019? Four years ago? Why is there no later search? The figure 1 – The search strategy needs to be presented clearer. What are the Publication dates – this should be presented? Why only three databases?
The procedure. The procedure of the review and the selections of publications should be part of the method section. The procedure to group the articles.
Results: What was the Definitions of socioeconomic status change in the articles? In years/age? The non-longitudinal design was excluded, but in your results, there are two cross-sectional studies – the reasons should be motivated. Downs and Black checklist were used – where is the quality score? The quality score should be part of table 1. The results are limited and perhaps re-organize the presentation in another way.
Discussion: Add up with new articles.
Reviewer 2 Report
The manuscript “A Review of the Relationship between Socioeconomic Status Change and Health” provides a helpful synthesis of published literature on SES change and health. Based on 11 included studies from a systematic screening process, the authors note eight main ways that SES change is measured and related to physical health, cardiovascular disease, mental health, and dental carries. Unsurprisingly, consistently high SES is related to better health, but changes from low to high SES are also related to improved health (and congruent with social theory). The paper identifies and synthesizes important literature but could benefit from several minor revisions that are outlined below.
Some accounting of context and the persistence of poverty such as described in Matthew Desmond’s work should be accounted for in the introduction. In addition, context is different across countries—e.g., patterns of racism and social exclusion--which should be described in the limitations section.
More detail on how the Downs and Black checklist was used should be discussed. In addition, what proportion of studies included were rated as excellent, good, or fair?
In figure 2, there is a column for additional records identified from other sources that says 0. I’m not sure this is necessary, unless gray literature was searched for and nothing was found—additional searching should be described in the text as well.
In table 1, the Alvarado study says multi-center under country. I looked it up and think it should state: Seven Latin American and Caribbean (LAC) cities.
In table 1, it appears several studies are labeled as cross sectional even though longitudinal was an inclusion criterion. I’m unclear why these would be included so they should be explained. I’d also like to see the timing of follow up data collection in the table.
It was notable that BMI was not correlated with SES change in general. Further discussion on this and references to studies pointing to limitations with BMI would be helpful.
Page 9, line 217 “nothing” should be “noting.”
In the conclusion and future research section, I’d recommend suggesting that future researchers test policies that promote improved economic wellbeing on health outcomes over time.
Round 2
Reviewer 1 Report
The authors have made a solid work with the revision. Not all. However references are missing. In the introduction and special the new added part. In the result section, the articles chosen for this study needs to be included, for example line 140-150 and 152-159. References needs to be added.
Author Response
Dear Reviewer,
Thank you very much for taking the time to review our manuscript. Your comments were very helpful in strengthening our manuscript.
In response to your second round of comments, we added references to the introduction section and the results, including the specified lines 140-150 and 153-160.